# Exploring Potential Biomarkers in Oesophageal Cancer: A Comprehensive Analysis

**DOI:** 10.3390/ijms25084253

**Published:** 2024-04-11

**Authors:** Adrianna Romanowicz, Marta Lukaszewicz-Zajac, Barbara Mroczko

**Affiliations:** 1Department of Biochemical Diagnostics, Medical University of Bialystok, ul. Waszyngtona 15a, 15-269 Bialystok, Poland; ada.ro@o2.pl (A.R.); mroczko@umb.edu.pl (B.M.); 2Department of Neurodegeneration Diagnostics, Medical University of Bialystok, ul. Waszyngtona 15a, 15-269 Bialystok, Poland

**Keywords:** biomarker, claudins, cytokines, oesophageal cancer, metalloproteinases

## Abstract

Oesophageal cancer (OC) is the sixth leading cause of cancer-related death worldwide. OC is highly aggressive, primarily due to its late stage of diagnosis and poor prognosis for patients’ survival. Therefore, the establishment of new biomarkers that will be measured with non-invasive techniques at low cost is a critical issue in improving the diagnosis of OC. In this review, we summarize several original studies concerning the potential significance of selected chemokines and their receptors, including inflammatory proteins such as interleukin-6 (IL-6) and C-reactive protein (CRP), hematopoietic growth factors (HGFs), claudins (CLDNs), matrix metalloproteinases (MMPs) and their tissue inhibitors (TIMPs), adamalysines (ADAMs), as well as DNA- and RNA-based biomarkers, in OC. The presented results indicate the significant correlation between the CXCL12, CXCR4, CXCL8/CXCR2, M-CSF, MMP-2, MMP-9 ADAM17, ADAMTS-6, and CLDN7 levels and tumor stage, as well as the clinicopathological parameters of OC, such as the presence of lymph node and/or distant metastases. CXCL12, CXCL8/CXCR2, IL-6, TIMP-2, ADAM9, and ADAMTS-6 were prognostic factors for the overall survival of OC patients. Furthermore, IL-6, CXCR4, CXCL8, and MMP-9 indicate higher diagnostic utility based on the area under the ROC curve (AUC) than well-established OC tumor markers, whereas CLDN18.2 can be used in novel targeted therapies for OC patients.

## 1. Introduction

Oesophageal cancer (OC) ranks as the ninth most prevalent cancer worldwide [1]. This malignancy is characterized by its aggressive nature, often evading detection until its advanced stages. OC leads to the death of over 50,000 people annually [2]. Therefore, it is also the sixth leading cause of cancer globally. Oesophageal squamous cell carcinoma (OSCC) and oesophageal adenocarcinoma (OAC) are the primary histological subtypes of this neoplasm. The incidence of OSCC (80%) predominates over OAC [3]. However, over the past decade, a rapid increase in the incidence of OAC has been observed, particularly in the western countries, thus, the number of new cases of OAC has increased up to 700% [4].

Oesophageal cancer risk factors are multifaceted, encompassing a complex interplay of genetic, environmental, and lifestyle elements. In general, these factors can be divided into categories such as modifiable and non-modifiable. In the first group, we can include tobacco, alcohol, and opium use, obesity, and gastro-oesophageal reflux disease (GERD). Chronic acid reflux, which causes inflammation, can lead to a condition called Barrett’s esophagus, which is a precursor to OC [5,6]. Furthermore, specific uncommon clinical syndromes, like tylosis palmaris et plantaris and Plummer–Vinson syndrome, are thought to elevate the risk of developing OSCC by causing oesophageal dysplasia and chronic inflammation [7]. The next group consists of non-modifiable risk factors comprising sex, age, and inherited genetic predisposition [6]. 

Considering the epidemiology of OC, the highest incidence rate occurs in Asia (the Asian oesophageal cancer belt) and in Africa (the African oesophageal cancer corridor), while the lowest rates of OC cases were noted for Central America and Western Africa [6]. Based on the 2020 Global Cancer Observatory, 69% of new OC cases were reported in males [1]. In middle-aged and elderly populations, the prevalence of this malignancy is greater and increases with age [8].

Although OC is often symptomless in its early stages, dysphagia, either alone or accompanied by unintentional weight loss, occurs as the most common presenting symptom [9]. Others include odynophagia, upper GI bleeding, hoarseness, and respiratory symptoms [10]. Understanding the differences between the types of OC allows us to more effectively customize prevention and early detection strategies. The diagnosis of OC is an extensive process comprising many steps. In clinical practice, imaging methods play an important role, such as computed tomography (CT), magnetic resonance imaging (MRI), positron emission tomography (PET), and endoscopic ultrasound scanning (EUS). Although clinical and demographic factors may raise suspicion about OC and its subtype, confirmation of the diagnosis typically requires an upper gastrointestinal endoscopy with biopsy [2,11,12]. Therefore, immunohistochemistry (IHC) is still used in confirmation of this malignant disease. Additionally, recent genetic studies of OC tumors identified distinctive mutational signatures and molecular patterns for each of the OSCC and OAC subtypes that may have potential therapeutic relevance [13,14]. Furthermore, based on the Cancer Genome Atlas, some authors have reported an integrated genomic landscape in OAC and OSCC, which can be utilized to identify potential therapeutic targets for OC subtypes [15]. In addition, the study of Valiullina et al. has indicated that chimeric antigen receptor (CAR-T) therapy can potentially be applied for the treatment of solid tumors, such as adenocarcinoma, and might estimate the risks of developing cytokine release syndrome [16]. Laboratory tests are also useful in the diagnosis of patients with this malignancy. The measurement of classical tumor markers, including squamous cell antigen (SCC-Ag) and carcinoembryonic antigen (CEA), is useful in the monitoring and follow-up of patients with this malignancy [17]. Although researchers have assessed various biochemical biomarkers for OC in the past decade, none have proven effective for early diagnosis due to their limited sensitivity and specificity at the early stages of the disease. Therefore, accessing the concentrations of these biomarkers is not valuable for the screening procedure. To enhance the treatment outcomes and reduce mortality rates among those with this malignancy, it is crucial to assess highly sensitive diagnostic tools, such as circulating blood markers or molecular biomarkers.

Numerous clinical and epidemiological studies have established a strong correlation between chronic inflammation and OC development [18]. In the tumor microenvironment (TME), a network of specific inflammatory agents engages a complex signaling process, potentially facilitating the movement of malignant cells through the stroma and contributing to the advancement of tumor progression [19,20]. Acute inflammation triggered by tumor-infiltrating leukocytes (TILs) may promote antitumor immunity, whereas prolonged chronic inflammation stimulates the synthesis of proinflammatory mediators and contributes to tumor initiation, promotion, and progression [21,22,23,24]. 

In this review, we summarize several original research papers that highlight the clinical significance of selected inflammatory mediators as potential biomarkers of OC in comparison to well-established classical tumor markers for OC. Little is known about the blood concentrations presented in the review of proteins that could be used as indicators of the prognosis and survival of OC patients. Therefore, additionally to the expression levels, we present the results of our previous studies, where we assessed the diagnostic and prognostic usefulness of serum chemokines and their receptors, including specific proteins such as interleukin-6 (IL-6) and C-reactive protein (CRP), hematopoietic growth factors (HGFs), claudins (CLDNs), matrix metalloproteinases (MMPs) and their tissue inhibitors (TIMPs), and adamalysines (ADAMs), in OC (Table 1, Figure 1)**.**

## 2. Specific Proteins as Potential Biomarkers of Oesophageal Cancer


### 2.1. Chemokines and Their Specific Receptors

Chemokines and their receptors have a crucial role in the initiation and growth of tumors. These proteins, which typically range in molecular mass from 8 to 10 kDa, are categorized into four structural groups, known as CXC, CX3C, CC, and C, based on the arrangement of the conserved cysteine residues crucial for their three-dimensional structure [46]. They can be classified into two functional categories: homeostatic and inflammatory chemokines [47,48]. CXC chemokines are particularly important in angiogenesis [49]. The impact of certain chemokines on tumorigenesis can vary, as some promote tumors while others act as tumor suppressors, depending on the specific cell type or receptor involved [50]. These molecules’ functions extend beyond the initially assigned roles of chemokines in attracting leukocytes to inflamed tissues. The functions of these molecules extend beyond the initially assigned roles of chemokines in attracting leukocytes to inflamed tissues. They encompass physiological processes like guiding sentinel cell populations to non-inflamed tissues, as well as influencing cellular activation, growth, adhesion, and phagocytosis [51]. Chemokines and cytokines cause changes in TME, surrounding tissues, and lymphoid organs, which promote tumor growth and prevent antitumor immune responses by activating immunosuppressive cells such as myeloid-derived suppressor cells (MDSCs), T regulatory cells (Tregs), tumor-associated macrophages (TAMs), and tumor-associated neutrophils (TANs). The recruitment of these cells is facilitated by chemokines and cytokines, leading to the creation of an immunosuppressive TME that inhibits CD4^+^/CD8^+^ T lymphocytes and natural killer (NK) cells from carrying out effective antitumor immune responses [25]. Some clinical investigations have demonstrated the potential role of CXC chemokines in the pathogenesis of various malignancies, including OC [26,52,53,54].

CXC chemokine ligand 12 (CXCL12), also known as stromal cell-derived factor-1 (SDF-1), pairs up with its C-X-C chemokine receptor type 4 (CXCR4) to relay information on leucocyte growth, adhesion, and migration. There is a growing body of evidence suggesting the presence of CXCL12 and its specific receptor CXCR4 in various tumor cells, indicating their potential involvement in different stages of tumor progression [52]. 

Based on the immunohistochemistry technique, investigators concluded that CXCL12 and its receptor levels depended on the amount of cancer cells [26]. Some authors have noted the expression of CXCL12 and CXCR4 in OC tissue, establishing significant correlations with invasion, angiogenesis, lymph node engagement, metastasis, and overall prognosis for OC patients [53]. Elevated CXCL12 levels may be related to a tendency of cancer cells to metastasize because it has been shown that an upregulation of CXCL12 and its specific receptor levels is associated with an increase in the advanced stages of OC, which might be a result of cancer’s ability to spread [26,53,54]. Moreover, the correlation between positive CXCL12 expression and nodal involvement, advanced tumor stage, and lymphatic invasion was evident [52]. Simultaneously, CXCR4 expression showed a significant increase in OSCC patients with lymph node metastasis and a T3 tumor stage compared to those without lymph node metastasis and patients with T1–T2 stages [26,27,28,52]. Furthermore, Sasaki et al. employed univariate analysis and found that OC patients with positive CXCL12 expression had significantly lower overall and disease-free survival rates compared to those with negative CXCL12 immunoreactivity. A total of 214 patients were selected based on specific criteria, such as their histopathological diagnosis of OSCC, absence of other tumors in other organs, absence of prior chemotherapy or radiotherapy treatment before surgery, and absence of endoscopic mucosal resection. Additionally, the patients were monitored in the postoperative period, allowing for the collection of data on disease recurrence. The authors concluded that CXCL12 expression serves as a crucial predictor for lymph node metastasis and an adverse prognosis in the survival of OSCC patients [52]. Research conducted by Gockel et al. indicated that no correlation was identified between CXCR4 expression and prognosis for those patients with OSCC and OAC [55]. Nevertheless, Wang et al. unveiled that a heightened CXCR4 expression served as a negative independent prognostic factor for the survival of OSCC patients [27]. 

In our previous studies, we investigated whether serum CXCL12 and its specific receptor (CXCR4) levels may be used as potential biochemical tumor markers for OC. We assessed the levels of CXCL12 and CXCR4 in the sera of 49 OC patients, considering the clinicopathological characteristics and comparing them with classical tumor markers (SCC-Ag and CEA) and markers of inflammation—CRP. The results indicated elevated CXCL12 and reduced CXCR4 levels in OC patients compared to the healthy controls. Moreover, concentrations of CRP were found to be higher in the OC patients than in the control group. It was reported that the elevated levels of CXCL12 and reduced expression of its specific receptor in OC may be due to an enhanced capacity of CXCR4 receptors to bind a greater amount of CXCL12 in cancer patients. The highest serum concentrations of this chemokine and its receptor were found in advanced stages of OC, similar to the SCC-Ag levels. CXCR4 exhibited the highest diagnostic sensitivity among all proteins tested, reaching 94% in the combined analysis with CXCL12, surpassing the diagnostic sensitivity achieved by combining CEA with SCC-Ag. This trend was similarly observed in the negative predictive value (NPV) for CXCR4 levels, which peaked among all analyzed proteins and rose to 80% when combined with CRP. Moreover, the area under the curve (AUC) for CXCR4 was similar to the AUC for CXCL12 and CEA and higher than that for SCC-Ag in the diagnosis of OC patients [42]. 

C-C motif chemokine ligand 5 (CCL5), known as RANTES (regulated upon activation, normal T cell expressed and presumably Secreted), is a member of the CC subfamily of chemokines and is expressed in memory T lymphocytes, macrophages, platelets, fibroblasts, dendritic cells, neurons, astrocytes, smooth muscle cells, and capillary endothelial cells. Apart from being a receptor for CCL3, CCR5 also serves as a receptor for other CC chemokines, including CCL4, CCL5, CCL8, and CCL3 like-1 (CCL3L1) [56,57]. This is an inflammatory chemokine that operates by attracting immune cells to sites of inflammation. 

OSCC cells originating from metastatic lymph nodes exhibit elevated CCL5 production compared to those from primary lesions. These cells express both CCR3 and CCR5 receptors, whereas normal oesophageal epithelial cells display low levels or lack chemokine and chemokine receptors [58]. Using small interfering RNA (siRNA) to reduce CCL5 expression results in a decrease in cancer cell proliferation, migration, and invasiveness, and an increase in apoptosis. In vitro studies show that Maraviroc, a CCR5 receptor antagonist, inhibits esophageal squamous cell carcinoma cell migration and invasion but has no effect on tumor growth [58]. Dunbar et al. have shown in their study that an elevated expression of CCL5 or CCR5 correlates with a poorer prognosis in low-grade oesophageal carcinomas [59].

C-X-C motif chemokine 8 (CXCL-8), also recognized as interleukin 8 (IL-8), is a member of the CXC chemokine subfamily with a unique arrangement of N-terminal cysteines. CXCL8 attaches to the cell surface of G protein-coupled receptors (CXCR1 and CXCR2) and is capable of activating various intracellular signaling pathways [60]. This chemokine has been confirmed as a neutrophil chemoattractant. However, it might contribute to the advancement of tumors by influencing the control of growth, proliferation, angiogenesis, and the survival of malignant cells, including OC [60,61]. Elevated expression of CXCL8 has been indicated in endothelial cells, infiltrating neutrophils, tumor-associated macrophages, and cancer cells [60]. 

The study of Ogura et al. assessed the expression of CXCL8 and its receptor (CXCR2) in OSCC tissue, as well as the association between the immunoreactivity of these proteins and the clinicopathological characteristics of cancer [29]. The subjects included 78 OSCC patients. The authors indicated that elevated CXCL8/CXCR2 expression significantly correlated with the higher depth of invasion, the presence of lymph node metastasis, as well as lymphatic and venous invasion [29]. Moreover, the co-expression of CXCL8 and CXCR2 was an independent predictive factor for the recurrence-free survival of OSCC patients, which was assessed using Cox’s hazards model. Consequently, the authors suggested that the overexpression of CXCL8 and CXCR2 could serve as a valuable diagnostic marker for guiding treatment decisions and predicting outcomes in OSCC patients [29]. Nguyen et al. utilized the quantitative reverse transcriptase real-time PCR (qRT-PCR) to examine the expression of inflammation-related genes, including CXCL8, in both OAC and non-tumor tissue from 93 patients. It was uncovered that the expression of the CXCL8 gene significantly escalated in OAC. Additionally, both tumor and non-tumor samples exhibited elevated CXCL8 expression levels, and these were associated with an unfavorable prognosis [62]. 

The serum concentration of CXCL8 in OC patients has been identified as a potential candidate for a tumor marker in both the diagnosis and progression of this malignancy. Analysis revealed markedly elevated CXCL8 levels in OC patients when compared to healthy individuals [43]. The study included 50 OC subjects and 26 volunteers. It was found that there are significant differences in the levels of CXCL8 and the degree of tumor invasion (T factor) among patients with OC and those with OSCC. Moreover, CXCL8 concentrations demonstrated a direct correlation with both the T factor and CRP levels. The diagnostic sensitivity, NPV, and the AUC of CXCL8 were higher than those of conventional tumor markers. These findings suggest that CXCL8 has the potential to aid in the diagnosis and monitoring of OC progression. 

### 2.2. Interleukin-6 (IL-6) and C-Reactive Protein (CRP) 

IL-6 and CRP play pivotal roles as potential biomarkers of OC [63,64]. Their intricate involvement in inflammatory processes and cellular signaling pathways underscores their significance in the pathogenesis of this malignancy. 

IL-6, a multifunctional proinflammatory cytokine, becomes activated in diverse cancer types such as breast, ovarian, and prostate cancer, renal cell carcinoma, along with multiple myeloma, leukemias, and lymphomas [63]. IL-6 constitutes a significant factor in a range of physiological processes, encompassing cell proliferation, migration, invasion, apoptosis, angiogenesis, and the growth and differentiation of cancer cells [64]. Through the activation of the JAK/STAT3 and PI3K/AKT signaling pathways, this cytokine can modulate cancer growth and dissemination within the body. Notably, elevated IL-6 levels have been observed in individuals with OC, correlating with disease progression and prognosis [65]. Furthermore, C-reactive protein is produced by hepatocytes in response to cytokines, especially IL-6, released by leukocytes within the tumor microenvironment [66,67]. Additionally, the hypothesis suggests that the presence of tumor cells, inducing tissue stress, can initiate an inflammatory response, serving as the stimulus for CRP synthesis [68]. Some researchers have confirmed that OC cells can produce IL-6, which stimulates CRP production [69]. Therefore, it was suggested that these inflammatory mediators may be potential candidates for biomarkers of OC.

The occurrence of IL-6 immunoreactivity was markedly higher in samples from OC compared to non-cancerous epithelium. This staining was significantly positively associated with the emergence of distant metastasis and lower rates of response to treatment [40]. Based on the findings by Chen et al., in the evaluation of clinical outcomes of 173 patients, a strong correlation was observed between positive IL-6 staining and an inadequate response to treatment, leading to a notably reduced survival [40]. In addition, elevated IL-6 serum levels were observed in OC patients, as indicated by the ELISA analysis [40]. In our previous studies, we analyzed the diagnostic and prognostic value of CRP and IL-6 levels measurement in OC in relation to different histological subtypes (OSCC and OAC) and compared them with classical tumor markers: CEA and SCC-Ag [17,30]. Serum concentrations of IL-6 and CRP in all the subgroups studied (OC, OSCC, and OAC patients) were significantly higher when compared with healthy subjects [17,30]. The study included 53 patients with OC and 90 healthy volunteers. Patients with any other concurrent malignant conditions, active infections, diabetes mellitus, or significant complications such as hepatic, cardiovascular, renal, or pulmonary diseases, as well as those who had received immunotherapy, chemotherapy, or radiation therapy before surgery, were excluded from the study. Moreover, the diagnostic sensitivity of IL-6 and CRP surpassed that of CEA and SCC-Ag to a significant extent in OC, similarly in OSCC and OAC patients. The IL-6 concentration was contingent on distant metastases and the survival of patients with OC, with elevated levels observed in individuals with more advanced tumor stages and nodal metastases, particularly in those diagnosed with OSCC. Moreover, the serum concentrations of CRP were significantly higher in OAC and OSCC patients with the presence of lymph node (N1) and distant metastases (M1) compared with the subjects without nodal involvement (N0) and distant metastases (M0). Additionally, the AUC for CRP was slightly lower than that for CEA but significantly higher than for SCC-Ag in all subgroups. In OSCC and OEC, the AUC for IL-6 exceeded that for both CEA and SCC-Ag. Interleukin-6 and C-reactive protein exhibit better diagnostic utility compared to traditional tumor markers in identifying OC, particularly in patients with OSCC [17,30]. Moreover, Huang et al. indicated a correlation between the elevated serum CRP levels and unfavorable overall survival (OS) in individuals with OC, with a hazard ratio (HR) of 1.40 and a 95% confidence interval (CI) of 1.25–1.57 [41]. These findings indicated that the measurement of serum IL-6 and CRP could potentially function as valuable biomarkers for OC. 

### 2.3. Hematopoietic Growth Factors (HGFs)

HGFs are cytokines that stimulate hematopoietic progenitor cells and may play a role in tumor regulation [70]. Clinical studies suggest abnormal HGF production in various malignancies, serving as growth factors and immune response indicators [71]. Moreover, these cytokines may be autocrine or paracrine growth factors [70]. Also, the receptors for various HGFs, including macrophage colony-stimulating factor (M-CSF), have been identified in non-hematopoietic tumor cell lines [70]. Authors have reported the presence of stem cell factor (SCF) mRNA in human OC cell lines [72]. 

In our previous studies, we concentrated on assessing the serum levels of various circulating hematopoietic cytokines, such as SCF and M-CSF, in OC patients in relation to the histological types (OSCC and OAC) and comparing these levels with classical tumor markers (CEA, SCC-Ag) and a healthy group [31,32]. In the investigations concerning serum SCF levels’ examination, 56 OC patients and 65 healthy controls were included, while 80 OC patients and 30 healthy subjects were employed in the study concerning the measurement of serum M-CSF concentrations [31,32]. The serum concentrations of M-CSF, CEA, and SCC-Ag were significantly higher in the OSCC and OC patients than in the healthy group [32]. Conversely, the SCF levels were notably reduced in the OC patients compared to the control group, while the CEA levels were higher in the OC patients than in the healthy control [31]. Moreover, the serum SCF concentrations were significantly higher in the OAC patients than in those with OSCC [31]. Additionally, the M-CSF levels in both the OC and OSCC groups tended to increase with the depth of tumor invasion, the presence of lymph nodes, and distance metastases [32]. However, we did not observe a similar correlation with SCF, as the levels of this cytokine decreased with the depth of the tumor [31]. When assessing the diagnostic significance of selected HGFs, the diagnostic sensitivity of SCF and M-CSF was higher than for the well-established tumor marker, CEA [31,32]. The combined analysis of M-CSF with SCC-Ag elevated their diagnostic sensitivity to 87%, surpassing that of the classical tumor markers in both analyzed groups (OC and OSCC) [32]. Our research indicates that M-CSF could serve as a promising tumor marker for OC, particularly in combination with SCC-Ag [31,32]. Rebernick et al. demonstrated with their univariate analysis a negative association between heightened levels of granulocyte-macrophage colony-stimulating factor (GM-CSF) in serum and overall survival (*p* = 0.040). The multivariate analysis determined that elevated serum GM-CSF was independently associated with lower survival, regardless of the crucial clinical factors (*p* = 0.001). The presented results indicate a negative correlation between the elevated GM-CSF levels and survival in OC. Importantly, these associations persist, irrespective of the pathological stage, and are consistent across various modalities and histologic subtypes [73]. Moreover, serum SCF concentrations were significantly higher in the OAC patients than in those with OSCC [31]. Additionally, the M-CSF levels in both the OC and OSCC groups tended to increase with the depth of tumor invasion, the presence of lymph nodes, and distant metastases [32]. However, we did not observe a similar correlation with SCF, as the levels of this cytokine decreased with the depth of the tumor [31]. When assessing the diagnostic significance of selected HGFs, the diagnostic sensitivity of SCF and M-CSF was higher than for the well-established tumor marker, CEA [31,32]. The combined analysis of M-CSF with SCC-Ag elevated their diagnostic sensitivity to 87%, surpassing that of the classical tumor markers in both analyzed groups (OC and OSCC) [32]. Our research indicates that M-CSF could serve as a promising tumor marker for OC, particularly in combination with SCC-Ag [31,32]. Rebernick et al. demonstrated in their univariate analysis a negative association between the heightened levels of GM-CSF in serum and overall survival (*p* = 0.040). The study was conducted on 47 patients with OC. The multivariate analysis determined that elevated serum GM-CSF was independently associated with lower survival, regardless of the crucial clinical factors (*p* = 0.001). The presented results indicate a negative correlation between the elevated GM-CSF levels and survival in OC. Importantly, these associations persist irrespective of the pathological stage and are consistent across various modalities and histologic subtypes [73].

### 2.4. Matrix Metalloproteinases (MMPs) and Their Tissue Inhibitors (TIMPs) 

MMPs are enzymes capable of breaking down proteins. They earn the title “metalloproteinases” due to their dependence on zinc and calcium for functionality [74]. MMPs are enzymes responsible for the degradation of extracellular matrix (ECM) components, which are essential for tissue remodeling and repair. TIMPs are a group of natural proteins that function as potent inhibitors of MMPs. In this sense, TIMPs act as guardians of the ECM integrity by regulating the activity of MMPs. By inhibiting the activity of MMPs, TIMPs prevent excessive ECM degradation, which could lead to tissue damage and disease progression. Hence, TIMPs are crucial players in maintaining the balance between ECM deposition and degradation in various physiological processes. These inhibitors are classified into four types: TIMP-1, TIMP-2, TIMP-3, and TIMP-4 [39,75]. Some clinical investigations suggested the usefulness of selected MMPs and their tissue inhibitors as potential biomarkers for OC [76]. Within the extensive range of around 30 MMP subtypes, MMP-2 and MMP-9 exhibit the most pronounced correlation with the development of this malignancy [76]. 

The authors identified the overexpression of MMP-2 and MMP-9 in OSCC using immunohistochemistry [33]. A total of 58 surgical specimens from patients with OSCC were collected. OSCC cells exhibited higher expression levels of MMP-2 (42.1%) and MMP-9 (60.3%) compared to the matched normal oesophageal tissues (22.9% for MMP-2 and 8.9% for MMP-9). Furthermore, MMP-2 expression in OSCC showed significant associations with tumor invasion depth, tumor-node-metastasis stages, and lymph node metastasis. The findings suggest that MMP-2 and MMP-9 may play crucial roles in the development of cancer, with MMP-2 serving as a potential biological marker for invasion and lymph node metastasis in OSCC [33]. 

Chen et al. conducted a study exploring the expression pattern of membrane-type 2 MMP (MT2-MMP) in OC tissues collected from 103 patients, examining its correlation with clinicopathological characteristics, neoangiogenesis, and postoperative prognoses. Their findings revealed that 85% of the total tumor sections exhibited positive MT2-MMP immunoreactions, while normal oesophageal tissues showed either none or very weak MT2-MMP staining. Furthermore, the intensity of MT2-MMP immunohistochemical staining was notably associated with angiogenesis in the tumor tissue and the size of the tumor [77]. Additionally, Murray et al. and El-Shahat et al. independently investigated the expression of MMP-9 in OC tissues [34,78]. In Murray’s study, 78% of the cases exhibited positive immunostaining, and the sensitivity increased with advancing tumor stages [34]. Meanwhile, El-Shahat reported a diagnostic sensitivity of 100% for MMP-9 immunostaining, particularly in stage IV of OC [78]. The research by Sharma et al. underscored the prognostic significance of TIMP-1 and TIMP-2 immunostaining in OSCC, particularly regarding invasion, tumor progression, and metastasis. In addition, patients with TIMP-2-negative carcinomas experienced significantly shorter disease-free survival compared to those with TIMP-2-positive tumors [35]. 

Some authors focused on exploring the diagnostic utility of the serum MMP-2 and MMP-9 levels in OC patients in relation to the clinicopathological features of cancer [44,76]. The serum concentrations of MMP-2, TIMP-2, CEA, and SCC were analyzed, demonstrating that OC patients had lower levels of MMP-2 and TIMP-2 than those in the healthy groups. The diagnostic sensitivity of TIMP-2 (57%) surpassed other biomarkers, especially in combination with SCC-Ag (70%). The AUC for TIMP-2 (0.8698) was larger than for CEA (AUC = 0.7958, *p* < 0.001), MMP-2 (AUC = 0.7405, *p* < 0.001), and for SCC-Ag (AUC = 0.5695, *p* = 0.3520) Additionally, the serum levels of MMP-2 and TIMP-2 tended to decrease in more advanced stages of cancer; therefore, the levels of both proteins did not show a significant correlation with the tumor stage. However, MMP-2 indicated a limited value for tumor staging and prognosis in OC, suggesting better usefulness of TIMP-2 as a potential tumor marker, particularly when combined with SCC-Ag [76]. Simultaneously, the serum levels of MMP-9 and SCC-Ag were significantly higher in the OSCC patients compared to the healthy subjects, particularly in advanced cancer cases. In this study, 63 patients diagnosed with OSCC were included alongside 30 healthy individuals. The diagnostic sensitivity of MMP-9 (75%) surpassed SCC-Ag (68%), and combining both markers demonstrated an increase of up to 97%. These results underscore the potential usefulness of MMP-9, especially in combination with SCC-Ag, for diagnosing OSCC [44]. Based on the presented research, selected MMPs and their tissue inhibitors may be valuable biomarkers in the progression and diagnosis of OC.

#### A Disintegrin and Metalloproteinase (ADAMs) Family 

ADAMs constitute a protein gene family characterized by two structural subgroups: the membrane-anchored ADAM and the variant featuring thrombospondin motifs, referred to as ADAMTS. These proteins are responsible for regulating the shedding of membrane-bound proteins, cytokines, growth factors, ligands, and receptors. What is unique about ADAMs is that they have both adhesive and proteolytic activities [79]. 

ADAM12 is part of the ADAMs family, a group of around 40 transmembrane and secreted metalloendopeptidases. Furthermore, ADAM12 exhibits two alternative splicing variants: a long form that is anchored to the membrane (ADAM12-L) and a short form released into the extracellular space (ADAM12-S) [80,81]. Some authors have revealed an increase in ADAM12-L expression in human OSCC tissues when compared to healthy tissues [82]. These findings have critical implications for OSCC patients, as elevated levels of ADAM12-L have been significantly linked to heightened rates of metastasis and poor prognosis [82]. 

As a crucial member of the ADAMs family, ADAM17, also known as tumor necrosis factor-α converting enzyme (TACE), plays a pivotal role as the primary secretase in liberating the soluble form of tumor necrosis factor-α from the plasma membrane [83]. The function of the ADAM17 protein involves the hydrolysis and release of precursor cell-surface proteins, thereby activating cell-surface molecules within the cell signaling pathway. This activation contributes to the modification of signal transmission in the tumor microenvironment, which is closely associated with tumorigenesis and tumor progression [84,85]. 

Liu et al. conducted research using specimens from 80 patients, comparing OSCC tissues and normal esophageal tissue using the immunohistochemical technique. The authors reported that the ADAM17 protein expression rates were found to be significantly higher in OSCC (66.25%) compared to its normal mucosa counterpart (6.25%) (*p* < 0.01) [86]. Additionally, there was a correlation between ADAM17 protein expression in oesophageal squamous cells and lymph node metastasis as well as the tumor stage (*p* < 0.05). However, no significant correlation was found between ADAM17 protein expression and gender, age, or histological grade [86].

Due to the function of ADAMs as a stress-triggered transcriptional repressor impacting plasminogen activator inhibitor-1 expression, ADAM9 might contribute to tumor vascularization [36]. Additionally, in OSCC, an increased expression of DNA methyltransferase 1 plays a role in promoting tumor growth by influencing ADAM9-mediated activation of the epidermal growth factor receptor (EGFR)-AKT signaling pathway. Some authors examined ADAM9 proteins by IHC staining in OSCC tissues obtained from 213 patients and demonstrated that the percentage of positive ADAM9 (61%) staining in advanced OSCC was significantly higher than in the early stages [36]. In patients with early-stage OSCC (stage I and II), those exhibiting positive ADAM9 staining experienced a shorter survival time compared to the ADAM9-negative group (*p* < 0.01) [36]. 

ADAMTS (a disintegrin-like and metalloprotease domain with thrombospondin), as zinc-dependent metzincins, are significant contributors to the adjustment of the extracellular matrix. Consequently, these proteins can cut through or engage with a diverse array of components in the extracellular matrix and regulatory factors, thereby influencing processes like cell adhesion, migration, proliferation, and angiogenesis [87]. 

Some authors concentrated on investigating the levels of ADAMTS-6 in patients with OSCC and estimated their prognostic significance [37]. Liu et al. indicated that the heightened ADAMTS-6 expression in tumor tissues correlated with an advanced tumor stage, nodal involvement, and recurrence. ADAMTS-6 expression was detected by IHC in 171 paraffin-embedded OSCC specimens. Kaplan–Meier survival curves demonstrated that increased ADAMTS-6 expression was associated with shorter overall survival (OS) (*p* = 0.001) and disease-free survival (DFS) (*p* = 0.002). The multivariate analysis confirmed that a high ADAMTS-6 expression independently predicted OSCC prognosis [37]. ADAMTS-6 expression showed significant correlations with Twist-related protein 1 (Twist-1) expression in both OSCC cancer cells (*p* = 0.007) and stromal cells (*p* < 0.001). Patients with OSCC expressing both ADAMTS-6 and Twist-1 exhibited markedly reduced OS and disease-free survival (DFS) rates compared to other patients [37]. Le Bras et al. have identified another enzyme, ADAMTS-1, playing a role in the invasion of oesophageal cells [76]. ADAMTS-1 is responsible for breaking down the proteoglycans aggrecan and versican and has demonstrated potential collagenase activity. Additionally, ADAMTS-1 is involved in activating stromal cells and has been observed to be upregulated by interleukin-1 beta (IL1β) while being downregulated by transforming growth factor β (TGFβ) in decidual stromal cells [88]. 

### 2.5. Claudins (CLDNs)

CLDNs are a group of transmembrane proteins that have a wide range of sizes and belong to a family of at least 27 members. Based on their sequence similarity, they are divided into classic claudins (CLDNs1–10, CLDN14, CLDN15, CLDN17, and CLDN19) and non-classic claudins (CLDNs11–13, CLDN16, CLDN18, and CLDNs20–24). A multi-gene family encodes these proteins, and several pairs of highly homologous CLDN genes are located close to each other in the human genome, e.g., CLDN8 and CLDN17 on chromosome 21 and CLDN3 and CLDN4 on chromosome 7 [89,90,91]. Claudins play a crucial role in tight junctions (TJs) by governing the permeability and polarity of the epithelial cells. The primary function of most CLDN family members is cell adhesion, providing a scaffold for cell migration, matrix remodeling, and proliferation [92]. These proteins are located in the apical region of the cell membrane and form a TJ complex that maintains cell polarity and adhesion. These proteins are in the apical region of the cell membrane and form a TJ complex that maintains cell polarity and adhesion. Consequently, CLDNs are essential for the proper functioning of the intercellular barrier maintained by TJs [93]. 

According to a study by Abu-Farsakh, CLDN2 expression was significantly higher in OAC, OSCC, and glandular lesions than in squamous cell carcinoma. The study did not find any significant relationship between the tissue levels of this protein and the age, gender, grade, stage, or survival time of patients with OAC and OSCC [94]. In the study by Gyõrffy et al., tissue blocks were taken from surgically removed tissues and biopsies, including OSCC (25 cases), BE (25 cases), and OAC (25 cases). They found that CLDN1 expression was higher in OSCC than in squamous epithelium. CLDN3 and CLDN4 were elevated in BE and OAC compared to foveolar epithelium, while CLDN2 was higher in OAC than in BE. The study suggests a link between Barrett’s esophagus and OAC in selected claudin patterns [95]. Usami et al. found that a decreased CLDN7 expression in OSCC was significantly associated with tumor invasion, stage, lymphatic vessel invasion, and lymph node metastasis [38]. In addition, the study by Moentenich et al., conducted on 485 patients with OAC, suggested that CLDN18.2 might be a promising therapeutic target. However, they did not establish a significant association between the CLDN 18.2 expression and clinical pathological features, such as the tumor stage and nodal involvement [45].

## 3. Molecular Biomarkers of Oesophageal Cancer (OC)

### 3.1. DNA-Based Biomarkers 

It is well established that the DNA methylation status can have significant impacts on an organism’s growth and development, as well as contribute to the development of tumors. In particular, DNA methylation tends to occur on regions in DNA that are rich in cytosine (C) and guanine (G), connected by a phosphate bond (CpG islands) located near the transcriptional regulatory regions. Researchers have shown a clear correlation between DNA methylation in these areas and the development of OC [96,97]. Zhou et al. indicated that the methylation of the *CDKN2A* promoter is more prevalent in patients with poorly differentiated tumors and advanced clinical stages, as opposed to those with well or moderately differentiated tumors and early-stage disease. *CDKN2A* promoter methylation in OC did not show any associations with the clinicopathological characteristics [98]. The diagnostic performance measures for OC based on *CDKN2A* methylation were sensitivity, specificity, and AUC, with values of 0.52 (95% CI, 0.44–0.59), 0.96 (95% CI, 0.93–0.98), and 0.83 (95% CI, 0.79–0.86), respectively [98]. The *p16* gene, referred to as the multiple tumor suppressor 1 (*MTS1*) gene, plays a direct role in overseeing the cell cycle, exerting a negative influence on both cell proliferation and division [99]. Its inactivation triggers the proliferation of malignant cells. The *p16* gene plays a significant role in the development of various tumors through gene deletion and mutation. It is crucial in clinical settings to identify changes in the *p16* gene, as it helps to assess the patient’s susceptibility to tumors and predict the prognosis of such conditions [100,101,102]. A study by Guo et al. showed that methylation of the *CDKN2A/p16INK4a* gene was predominantly observed in OSCC, accounting for around 52% of the cases [103]. Li et al. reported that the pathogenesis of OSCC was related to hypermethylation of several tumor-related genes, such as *RAR-β*, *p16*, *DAPK*, and *CDH1*, through heightened *DNMT3b* expression [104]. Li et al. reported that the pathogenesis of OSCC involves hypermethylation of multiple tumor-related genes, such as *RAR-β*, *p16*, *DAPK*, *RASSF1A*, and *CDH1*, which is mediated by an increase in *DNMT3b* expression. Serum samples from 45 patients and 15 healthy individuals were analyzed for the methylation status of five genes. It was found that hypermethylation of cell-free serum DNA was prevalent among OSCC patients. Additionally, diagnostic accuracy significantly improved when assessing the methylation of multiple genes (*RAR-β, DAPK, CDH1, p16*, and *RASSF1A*) together, with a resulting ROC AUC of 0.911, a sensitivity of 82.2%, and a specificity of 100% [104]. Additionally, methylenetetrahydrofolate reductase (MTHFR) is the rate-limiting enzyme that regulates the metabolism of folate and methionine [105,106]. A study by Song et al. indicated that genetic variations in the *MTHFR* gene could be a predisposing factor for OSCC in both an experimental group of 240 OC patients and a control group of 360 healthy individuals [105]. 

Moreover, *TP53* mutations primarily occur during the progression of nondysplastic Barrett’s esophagus to high-grade dysplasia (HGD), meaning that *TP53* gene mutations are already present in most cases with HGD as well as in patients diagnosed with OC [106]. A study investigating the prognostic significance of *TP53* mutations in OAC found that 47% of the analyzed tumors harbored *TP53* mutations, primarily G:C to A:T transitions at CpG dinucleotides. These mutations correlated with the increased expression of p53 protein, tumor differentiation, and notably diminished postoperative survival following surgical resection of OAC [107]. Moreover, mutations causing the loss of *SMAD4* function are primarily detected in the disease’s advanced stages, indicating that *SMAD4* mutations or deletions emerge as significant drivers in the progression of OAC [108]. Hence, alterations in the *TP53* and *SMAD4* genes are expected to play a critical role in the progression to OAC. Significantly, patients with Smad4-negative tumors had a shorter time to recurrence (TTR) (*p* = 0.007) and worse overall survival (*p* = 0.011) [109]. In a separate cohort of five locoregional recurrences and 43 metastatic esophageal adenocarcinomas, Smad4 was evaluated due to its correlation with postoperative locoregional and/or distant metastases. When compared to primary tumors, metastatic disease showed a higher prevalence of Smad4 loss (44% vs. 10%). 

*EYA4*, *PAX1*, *SOX1*, *ZNF582*, and *Polι* were all identified in association with DNA methylation in OC and might be biomarkers of abnormal DNA methylation in oesophageal carcinoma (Table 2).

It was proved that DNA methylation has great utility for early OSCC diagnosis. Authors using the ctDNA plasma methylation platform revealed the feasibility of OC detection up to four years before the clinical diagnostic [114]. In the investigation based on methylation patterns in cfDNA that included 6689 participants (2482 with cancer and 4207 without cancer), the specificity of cancer detection was nearly 100%, and the sensitivity was 67.3% in 12 types of cancer, including OSCC [115]. The cfDNA template is found in specific body fluids at very low concentrations, which may lead to the low sensitivity of detection. A growing body of evidence suggests that liquid biopsy has recently emerged as a new non-invasive technique of detecting blood circulating biomarkers for OSCC [116], e.g., detection of circulating tumor cells (CTCs) and circulating tumor DNA (ctDNA) [117,118]. Authors have reported that a liquid biopsy is a novel tool for diagnosis, prognostic stratification, and personalized therapy, while liquid biopsy biomarkers for OSCC could be useful in early detection and may provide prognosis information [119].

### 3.2. RNA-Based Biomarkers

Many research studies have investigated the predictive impact of non-coding RNAs on OC, which are crucial in the development, infiltration, and metastasis of tumor cells [119]. There are two primary types of non-coding RNAs (ncRNAs): structural ncRNAs, which mainly consist of rRNA and tRNA, and regulatory ncRNAs [120,121]. According to a recent comprehensive prospective investigation, identifying tRNA-derived small RNAs (tsRNA) in the salivary exosomes of patients effectively distinguishes those with OC from healthy individuals, achieving a specificity of 94.2% [122]. This discovery serves as a promising novel prognostic indicator for oesophageal cancer. Additionally, microRNA (miRNA) is a group of single-chain non-coding RNAs that have been found to be evolutionarily conserved. They are known to participate in various physiological processes such as cell differentiation, proliferation, metabolism, and apoptosis [123]. A prognostic risk model for oesophageal adenocarcinoma, utilizing miR-4521, miR-3682-3p, and miR-1269a, has established a significant correlation between the miRNA target genes and immune infiltration, tumor microenvironment, cancer stemness properties, and tumor mutational load in oesophageal adenocarcinoma [124]. Apart from miRNAs, long non-coding RNAs (lncRNAs) also have the potential to be used for predicting and treating oesophageal cancer. For instance, Xue et al. conducted a transcriptome analysis and discovered that LINC00680 was highly expressed in oesophageal cancer and had a significant correlation with the tumor volume, stage, and prognosis of the disease [125]. Additional investigations demonstrated that suppressing LINC00680 expression led to a partial reduction in the proliferative properties of OC, both in vitro and in vivo. The primary mechanism of action was found to be that LINC00680 acts as a competing endogenous RNA (ceRNA) that can sponge miR-423-5p, which in turn regulates the expression of p21-activated kinase 6 (PAK6) in oesophageal cancer cells [125]. Other prognostic factors related to OC encompass LncRNA TUG1, LncRNA FAM83A-AS1, miR-2053, miR-493, and LncRNA RPL34-AS1 (Table 3).

## 4. Conclusions

Presently, the annual toll of cancer-related fatalities escalates steadily. Oesophageal cancer, a prominent digestive system malignancy, poses an ongoing challenge for humanity. The research on the biomarkers in this malignancy holds significant promise for early detection and personalized treatment strategies. Thus, the establishment of new biomarkers that will be measured with non-invasive techniques at a low cost is a critical issue in improving the diagnosis of OC. 

In this work, we demonstrated the ability of selected proteins as promising blood biomarkers of OC. Our review shows the potential usefulness of novel biomarkers and compares their significance with well-established tumor markers such as CEA and SCC-Ag. The findings of the presented papers indicate the significant association between the CXCL12, CXCR4, CXCL8/CXCR2, M-CSF, SCF, MMP-2, MMP-9, ADAM17, ADAM9, ADAMTS-6, and CLDN7 levels and tumor stages of OC. The studies revealed that the levels of CXCL12, CXCR4, CXCL8/CXCR2, IL-6, M-CSF, MMP-2, MMP-9 ADAM17, ADAMTS-6, and CLDN7 significantly correlated with the clinicopathological parameters of OC, such as the presence of lymph node and/or distant metastases. Moreover, CXCL12, CXCL8/CXCR2, IL-6, TIMP-2, ADAM9, and ADAMTS-6 were prognostic factors for the overall survival of OC patients. Furthermore, IL-6, CXCR4, CXCL8, and MMP-9 indicate higher diagnostic utility based on the AUC than the well-established tumor markers, such as CEA and SCC-Ag in OC patients. The presented results expand on the recent knowledge of the therapeutic value of CLDN18.2 for OC patients. The current findings add evidence, suggesting that further investigations concerning the assessment of new blood biomarkers are sorely needed to improve the overall survival of patients with this deadly malignancy. However, the lack of investigations concerning the measurement of blood concentrations of these molecules in large groups of OC patients is the main reason why further studies need to be performed. 

## Figures and Tables

**Figure 1 ijms-25-04253-f001:**
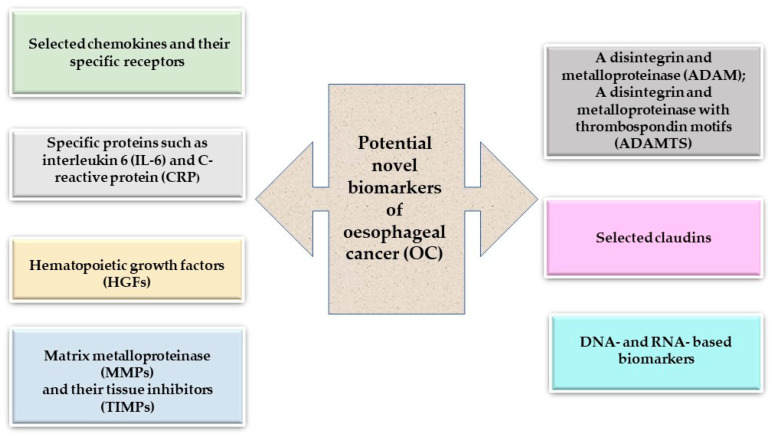
Potential novel biomarkers for oesophageal cancer (OC).

**Table 1 ijms-25-04253-t001:** The significance of analyzed proteins in oesophageal cancer (OC).

Clinical Significance	Protein	References
The correlation between protein levels and tumor stage	CXCL12, CXCR4, CXCL8/CXCR2, CRP, M-CSF, SCF, MMP-2, MMP-9, ADAM17, ADAM9, ADAMTS-6, CLDN7	[17,25,26,27,28,29,30,31,32,33,34,35,36,37,38]
The correlation between protein levels and the presence of lymph node metastases	CXCL12, CXCR4, CXCL8/CXCR2, CRP, IL-6, M-CSF, MMP-2, MMP-9, ADAM17, ADAMTS-6, CLDN7	[25,26,27,28,29,32,33,38,39]
The correlation between protein levels and the presence of distant metastases	IL-6, CRP, M-CSF	[17,30,32,40]
The correlation between protein levels and survival of OC patients	CXCL12, CXCL8/CXCR2, IL-6, TIMP-2, ADAM9, ADAMTS-6	[25,29,35,36,37,41]
Diagnosis of OC patients	CXCR4, CXCL8, IL-6, MMP-9	[17,42,43,44]
Novel targeted therapies for OC patients	CLDN18.2	[45]

**Table 2 ijms-25-04253-t002:** Selected genes as potential biomarkers in oesophageal cancer (OC).

Genes	Functions	Methylation Status and References
*EYA4*	Engaged in the regulation of apoptosis, innate immunity, DNA damage repair, and angiogenesis	Hypermethylation status [110,111]
*PAX1*	Engaged in the control of transcription, DNA-dependent processes, and promoter development	Hypermethylation status [112]
*SOX1*	Engaged in the establishment and upkeep of chromatin structure, regulation of transcription, and is DNA-dependent	Hypermethylation status [112]
*ZNF582*	Engaged in transcriptional regulation	Hypermethylation status [112]
*Polι*	Engaged in the translation process and synthesis of DNA	Hypomethylation state [113]

**Table 3 ijms-25-04253-t003:** Biomarkers of ncRNAs in oesophageal cancer.

Biomarkers of ncRNAs	Mechanism	References
LncRNA TUG 1	Control of the miR-1294/PLK1 pathway, a pivotal oncogenic mechanism in oesophageal cancer.	[126,127]
LncRNA FAM83A-AS1	Intensification of malignant advancement by suppressing the activity of miR-495-3p.	[128]
miR-2053	Elevation of kinesin family member 3C (KIF3C) expression and initiation of the PI3K/AKT signaling pathway are implicated in cell proliferation, apoptosis, migration, and invasion.	[129]
miR-493	Suppresses transcription factor Jun (c-JUN) and p-PI3K/p-AKT activity, enhances p21	[130]
LncRNA RPL34-AS1	Upregulation of RPL34-AS1, suppressed cell proliferation, colony formation, invasion, and migration in vitro, overexpression of RPL34-AS1, and inhibited tumor growth in vivo	[131,132]

## Data Availability

No new data were created or analyzed in this study. Data sharing is not applicable to this article.

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
