# Peer review of "Exploring Potential Biomarkers in Oesophageal Cancer: A Comprehensive Analysis"

_ijms, 2024, doi:10.3390/ijms25084253_

Round 1

Reviewer 1 Report

Comments and Suggestions for Authors

Adrianna Romanowicz and co-authors present a quality and well-written review manuscript focused on exploring potential biomarkers in oesophageal cancer.

Authors summarized several original studies concerning the potential significance of selected chemokines and their receptors, inflammatory proteins such as interleuin-6 (IL-6) and C-reactive protein (CRP), hematopoietic growth factors (HGFs), claudins (CLDNs), matrix metalloproteinases (MMPs) and their tissue inhibitors (TIMPs), adamalysins (ADAMs), as well as DNA- and RNA-based as biomarkers of OC. 

Authors suggest that presented results indicate the correlation between CXCL12, CXCR4, CXCL8/CXCR2, M-CSF, SCF, MMP-2, MMP-9, ADAM17, ADAM9, ADAMTS-6, CLDN7 levels and tumor stage of OC. The studies revealed that CXCL12, CXCR4, CXCL8/CXCR2, IL-6, M-CSF, MMP-2, MMP-9 ADAM17, ADAMTS-6 CLDN7 levels correlated with clinicopathological parameters of OC, such as the presence of lymph node and/or distant metastases. CXCL12, CXCL8/CXCR2, IL-6, TIMP-2, ADAM9 and ADAMTS-6 were prognostic factors for overall survival in OC patients. Furthermore, IL-6, CXCR4, CXCL8 and MMP-9 indicate higher diagnostic utility than well-established OC tumor markers, whereas CLDN18.2 can be used in novel targeted therapies for OC patients. 

Authors cover such aspects as:

- Chemokines and their specific receptors

- Interleukin-6 and C-Reactive Protein 

- Hematopoietic growth factors 

- Matrix metalloproteinases and their tissue inhibitors 

- Claudins 

- DNA-based biomarkers

- RNA-based biomarkers

Finally, authors conclude that they showed the potential usefulness of novel biomarkers and compares their significance with classical tumor markers such as CEA and SCC-Ag. The findings of the presented papers indicate the correlation between CXCL12, CXCR4, CXCL8/CXCR2, M-CSF, SCF, MMP-2, MMP-9, ADAM17, ADAM9, ADAMTS-6, CLDN7 levels and tumor stage of OC. The studies revealed that the levels of CXCL12, CXCR4, CXCL8/CXCR2, IL-6, M-CSF, MMP-2, MMP-9 ADAM17, ADAMTS-6 CLDN7 correlated with clinicopathological parameters of OC, such as the presence of lymph node and/or distant metastases.

==============================

Overall, the manuscript is highly valuable for the scientific community and should be accepted for publication after the corrections are made.

Other comments:

1) Please check for typos throughout the manuscript.

2) Please improve figures/tables where appropriate.

3) Authors are kindly encouraged to cite the following article that describes novel therapeutic approaches for the treatment of solid tumors, including carcinomas. DOI: 10.3390/biomedicines11020626

Author Response

Reviewer 1

Overall, the manuscript is highly valuable for the scientific community and should be accepted for publication after the corrections are made.

DETAILED ANSWERS TO THE REVIEWERS’ COMMENTS

  1. Please check for typos throughout the manuscript.

Authors' Responses to Reviewer's Comments

We are very grateful for your kind review.

Spelling through the whole body of the manuscript has been checked and corrected, as it was recommended.

  1. Please improve figures/tables where appropriate.

Authors' Responses to Reviewer's Comments

Figures and tables have been improved, in the revised manuscript, according to the Reviewer 1 suggestion.

  1. Authors are kindly encouraged to cite the following article that describes novel therapeutic approaches for the treatment of solid tumors, including carcinomas. DOI: 10.3390/biomedicines11020626

Authors' Responses to Reviewer's Comments

New reference: “Valiullina AK, Zmievskaya EA, Ganeeva IA, Zhuravleva MN, Garanina EE, Rizvanov AA, Petukhov AV, Bulatov ER. Evaluation of CAR-T Cells' Cytotoxicity against Modified Solid Tumor Cell Lines. Biomedicines. 2023 Feb 19;11(2):626” concerning the novel therapeutic approaches for the treatment of solid tumors has been added in the new version of the paper, as it was recommended (page 2, lines 83-86; page 15, lines 1455-1456).

Reviewer 2 Report

Comments and Suggestions for Authors

In this study, Authors provided analyses of the role of specific chemokines and their receptors, particularly conducing on CXCL12/CXCR4 and CXL8/CXCR2, as well as the potential relationship of IL6/CRP and OC, Authors also discussed the role of HGFs in stimulating hematopoietic progenitor cells and regulating tumor proliferation, and highlighted the roles of MMPs and TIMPs in tissue damage and disease progression. Additionally, the significance of CLDNs in OC was explored, highlighting their role in tight junctions and cell adhesion. DNA methylation and alterations, and RNA-based biomarkers, including tsRNAs, miRNAs and lncRNAs were reviewed due to their involvement in tumorigenesis and regulation. It is an informative work for offering key aspects of OC. However, expanding the scope of the study could attract a wider audience.

1.     Several studies have investigated mutations in genes such as TP53, CDKN2A, and SMAD4 associated with EC, which was not addressed in this study.

2.     Regarding chemokine receptor expression, CCL5/CCR5 have also been found to be associated with poor prognosis and metastasis.

3.     Since chemokines and their receptors play a crucial role in OC, the modulation of immune cell recruitment and function through chemokines/receptors should be included.

4.     In addition, adding studies of liquid biopsies for detecting genetic and epigenetic alterations might highlight the unique contribution of this study, since previous reviews didn’t cover this aspect.

Comments on the Quality of English Language

It is ok.

Author Response

Reviewer 2

DETAILED ANSWERS TO THE REVIEWERS’ COMMENTS

  1. Several studies have investigated mutations in genes such as TP53, CDKN2A, and SMAD4 associated with EC, which was not addressed in this study.

Authors' Responses to Reviewer's Comments

We are very grateful for your kind review and suggestions.

The studies investigated mutations in genes such as TP53, CDKN2A, and SMAD4 associated with EC have been added, as it was recommended (pages 11-12, lines 1048-1055; 1062; 1065-1073; 1078-1149). In addition, new references have been added in the revised version of the paper (page 18, lines: 1613; 1628-1639).

  1. Regarding chemokine receptor expression, CCL5/CCR5 have also been found to be associated with poor prognosis and metastasis.

Authors' Responses to Reviewer's Comments

Additional information concerning the association between chemokine receptor expression, CCL5/CCR5 and poor prognosis/metastasis of EC has been added in the in the revised manuscript, according to the Reviewer 2 suggestion (page 5, lines 394-411).  In addition, new references have been added in the revised version of the paper (page 16, lines 1492-1499).

  1. Since chemokines and their receptors play a crucial role in OC, the modulation of immune cell recruitment and function through chemokines/receptors should be included.

Authors' Responses to Reviewer's Comments

The information concerning the modulation of immune cell recruitment and function through chemokines/receptors has been added, as it was recommended (page 4, lines 281-282; 286-293). In addition, new reference has been added in the revised version of paper (page 16, lines 1473).

  1. In addition, adding studies of liquid biopsies for detecting genetic and epigenetic alterations might highlight the unique contribution of this study, since previous reviews didn’t cover this aspect.

Authors' Responses to Reviewer's Comments

Thank you for interesting suggestion. The information concerning the significance of liquid biopsies for detecting genetic and epigenetic alterations has been added in the revised version of the manuscript, as it was recommended (pages 12-13, lines 1155-1229). In addition, new references have been added in the new version of the paper (page 19, lines 1648-1665).

Reviewer 3 Report

Comments and Suggestions for Authors

In this Review, the Authors focused on the characterization of new biomarkers of oesophagal cancer (OC) that can be evaluated with non-invasive techniques. They summarized investigations related to potential significance of chemokines , hematopoietic growth factors, claudins, matrix metalloproteinases, adamalysins, as well as DNA- and RNA-based as biomarkers of OC.

This Review is well written and is very interesting to read. It can be published.

It would be good only to underline principal novelty compared to other reviews and outline perspectives in conclusion.

Author Response

Reviewer 3

DETAILED ANSWERS TO THE REVIEWERS’ COMMENTS

  1. It would be good only to underline principal novelty compared to other reviews and outline perspectives in conclusion.

Authors' Responses to Reviewer's Comments

We are very grateful for your kind review and suggestions.

The manuscript has been modified to underline principal novelty of paper, according to the Reviewer 3 suggestion (pages 11-13, lines 1048-1055; 1065-1073; 1078-1149; 1155-1229). Moreover, the conclusion section has been modified, as it was recommended (page 14, lines 1374-1391). New references concerning the novel liquid biopsies biomarkers as well as new therapeutic techniques have been added in the revised version of the manuscript (page 18-19, lines: 1628-1639; 1648-1665).

Reviewer 4 Report

Comments and Suggestions for Authors

Congratulation for your work and your efforts to support with your data advanced diagnostic options for oesophageal carcinoma patients. We should always encourage investigators who orchestrate efforts and ideas to provide the scientific community with new and significant information. I personally acknowledge that you have contributed laboriously to define your objectives but I have serious worries about the methodology followed and the interpretation of your results. Noteworthy the extensive analysis of your data provided valuable information but however there are some points that I would like to comment and your response will be definitely appreciated.

Comment1:This is not a systematic review and has to be mentioned in the abstract body and in methodology chapter

Comment2: Even it is not a systematic review the total number of patients that participated in the human studies must be shown in the manuscript

Commen3: You should describe the knowledge gained in the research field for each organized team of possible biomarkers separately for the in vitro and in vivo studies. Even more if the results in the studies were normalized statistically for confounders such as BMI, age, disease stage, secondary metastases and the presence of other chronic diseases while their pathophysiology could explain activation of the inflammatory status.

Comment 4: It would be interesting to know the behavior of the proposed biochemical biomarkers in the presence of their inhibitors if there are any relevant studies.

Comment 5: In case of candidate genes then in orders to study their combined burden, genetic risk scores has been developed otherwise we only have some observations presented.

Comment 6: In order some biochemical genetic and epigenetic parameters will be proposed we need sensitivity and specificity statistical evaluations to accompany the conclusion, otherwise we only have observations.

Comment 7: Please state any possible strengths or limitations with the methodology followed to support your conclusions

Comment 8: The conclusions and output of this study need to be more detailed or interpreted. What are the clinical outcomes of this study that will be helpful ? Please discuss and explain.

Comments on the Quality of English Language

Moderate editing from a natively English speaking reviewer is recommended

Author Response

Reviewer 4

DETAILED ANSWERS TO THE REVIEWERS’ COMMENTS

  1. This is not a systematic review and has to be mentioned in the abstract body and in methodology chapter
  2. Even it is not a systematic review the total number of patients that participated in the human studies must be shown in the manuscript

Authors' Responses to Reviewer's Comments

We are very grateful for your kind review and interesting suggestions.

The abstract as well as the conclusion sections have been modified (page 1, lines 18-24; page 14, lines 1376-1393). In addition, in the abstract as well as in the body of the manuscript there is no statement that our review is a systematic review: “In this review we summarize several original studies concerning the potential significance of selected chemokines and their receptors…” (page 1, lines 13-14). Moreover, the total number of patients that participated in most human studies described in the review has been added in the new version of the paper, as it was recommended. 

  1. You should describe the knowledge gained in the research field for each organized team of possible biomarkers separately for the in vitro and in vivo studies. Even more if the results in the studies were normalized statistically for confounders such as BMI, age, disease stage, secondary metastases and the presence of other chronic diseases while their pathophysiology could explain activation of the inflammatory status.

Authors' Responses to Reviewer's Comments

The knowledge gained in the research field for each group of possible biomarkers has been organized separately for the in vitro and in vivo studies (pages 3-11). In addition, the information concerning the presence of other inflammatory diseases in participants of described studies has been added in the revised manuscript, according to the Reviewer 4 suggestion (page 6, lines 495-499).

  1. It would be interesting to know the behavior of the proposed biochemical biomarkers in the presence of their inhibitors if there are any relevant studies.

Authors' Responses to Reviewer's Comments

The information concerning the behavior of the proposed biochemical biomarkers such as matrix metalloproteinases and their tissue inhibitors as well as the role of selected drugs as inhibitors of analyzed biomarkers has been added in the new version of the paper, as it was recommended (page  5 and 11, lines 402-411; 1038-1039). In addition, new references have been added in the revised version of the paper (page 16, lines 1496-1500).

  1. In case of candidate genes then in orders to study their combined burden, genetic risk scores has been developed otherwise we only have some observations presented.
  2. In order some biochemical genetic and epigenetic parameters will be proposed we need sensitivity and specificity statistical evaluations to accompany the conclusion, otherwise we only have observations.

Authors' Responses to Reviewer's Comments

Thank you for interesting suggestion.

The sensitivity and specificity statistical evaluations have been added, as it was recommended (pages 11-12, lines 1049-1056; 1066-1074). In addition, the references concerning novel, non-invasive technique for genetic and epigenetic parameters have been added in the revised version of the manuscript, according to the Reviewer 1,2 and 4 (pages 11-13, lines 1049-1056; 1066-1074; 1156-1230).

  1. Please state any possible strengths or limitations with the methodology followed to support your conclusions.
  2. The conclusions and output of this study need to be more detailed or interpreted. What are the clinical outcomes of this study that will be helpful ? Please discuss and explain.

Authors' Responses to Reviewer's Comments

Thank you very much for interesting comment.

The conclusion section has been modified according to Reviewer 3 and 4 suggestion.The conclusion section has been formulated based on statistically significant results presented in original papers by other authors and described in this review. Currently, there are only few data assessing the concentrations of selected ADAMs and claudins in the blood of patients OC using ELISA and Luminex method. However, actually our research team is conducting analysis of the serum concentration of ADAM15 and ADAM12 as well as serum concentrations of CLDN1,-2, -3, -4, -5, -6, -8, -18 in CRC patients in comparison to well-established, classical tumor markers of GI tumors such as carcinoembryonic antigen (CEA) and cancer antigen 19-9 (CA 19-9). The preliminary research results are promising, therefore we plan to evaluate the diagnostic criteria such as diagnostic sensitivity and specificity, predictive value of true positive and negative results and diagnostic accuracy of selected serum biomarkers in CRC patients. Based on those limitations the conclusion section has been modified, as it was recommended (page 14, lines 1376-1393).

  1. Moderate editing from a natively English speaking reviewer is recommended

Authors' Responses to Reviewer's Comments

The paper has been corrected by a English language translator, as it was recommended.

Round 2

Reviewer 2 Report

Comments and Suggestions for Authors

My concerns were addressed. Thanks!

Comments on the Quality of English Language

It is OK.